# Structured Recognition for Generative Models with Explaining Away

**Changmin Yu**[1,2]    **Hugo Soulat**[2]    **Neil Burgess**[1]    **Maneesh Sahani**[2]

[1]Institute of Cognitive Neuroscience;    [2]Gatsby Computational Neuroscience Unit;
UCL, London, United Kingdom
{changmin.yu.19; hugo.soulat.19; n.burgess}@ucl.ac.uk;
maneesh@gatsby.ucl.ac.uk

## Abstract

A key goal of unsupervised learning is to go beyond density estimation and sample generation to reveal the structure inherent within observed data. Such structure can be expressed in the pattern of interactions between explanatory latent variables captured through a probabilistic graphical model. Although the learning of structured graphical models has a long history, much recent work in unsupervised modelling has instead emphasised flexible deep-network-based generation, either transforming independent latent generators to model complex data or assuming that distinct observed variables are derived from different latent nodes. Here, we extend amortised variational inference to incorporate structured factors over multiple variables, able to capture the observation-induced posterior dependence between latents that results from "explaining away" and thus allow complex observations to depend on multiple nodes of a structured graph. We show that appropriately parametrised factors can be combined efficiently with variational message passing in rich graphical structures. We instantiate the framework in nonlinear Gaussian Process Factor Analysis, evaluating the structured recognition framework using synthetic data from known generative processes. We fit the GPFA model to high-dimensional neural spike data from the hippocampus of freely moving rodents, where the model successfully identifies latent signals that correlate with behavioural covariates.

## 1   Introduction

A central challenge of unsupervised learning is to identify and model patterns of statistical dependence in high-dimensional data. One approach to this challenge exploits latent-variable generative models, which capture the statistical structure of data through the modelled influence of the latent variables on observations, and through interactions between the latents themselves. Recent developments in deep learning have enabled deep generative models (DGM), with remarkable success in density estimation and high-fidelity image or text generation [1–5]. However, much of the DGM development has emphasised the expressiveness and accuracy of the latent-to-observation generative process, with the latents themselves often assumed independent *a priori*.

Variational learning requires inferring or approximating the posterior distribution of the latents conditioned on observations [6]. In a variational autoencoder [VAE; 1, 2], inference for a DGM is *amortised* by training a recognition network to return parameters of the variational posterior. The structured variational autoencoder [SVAE; 7] connects the DGM framework with richer models that include structured prior dependence amongst the latent variables. In an SVAE the structured latent prior is combined with a DGM link between each latent and a corresponding observation. Variational inference is amortised using recognition networks that return parameters of a factor associated with each link, which are combined with the structured prior to obtain the full posterior. However, in

many generative models of interest, observations depend on more than one latent variable. Such interactions induce joint potentials in the likelihood, coupling latents in the posterior even if they are independent *a priori*, a phenomenon sometimes called "explaining away" [8]. To capture such observation-induced dependence in the DGM setting we develop the *structured recognition VAE* (SRVAE), where recognition potentials that incorporate joint factors induced by the observations are learnt in a (structured) variational autoencoding framework.

We instantiate the SRVAE framework for a range of graphical structures, including non-linear latent Gaussian process (GP) models. In the latter case, we bring together the GP-prior VAE [9, 10], and (sparse) Gaussian process factor analysis (GPFA) [11, 12] models, yielding a novel autoencoding model (SR-nlGPFA). Experiments show that SR-nlGPFA outperforms alternatives that lack structured recognition. We apply SR-nlGPFA to data collected from a population of neurons in the hippocampal complex, and show that the unsupervised approach automatically captures dimensions underlying neural firing that reflect relevant behavioural correlates.

## 2 Background

### 2.1 Variational Inference and Structured Variational Autoencoders

Consider a general generative process with latent variables $z$ and observations $y$,

$$z \sim p(z|\theta), \quad y \sim p(y|z; \gamma),$$

where $\theta$ and $\gamma$ are the parameters of the prior and conditional likelihood distributions, respectively (and we write $\Theta = (\theta, \gamma)$). We assume deterministic $\theta$ for simplicity, but distributions over $\theta$ (with conjugate hyperpriors) can be incorporated by variational Bayes (VB) [13]. The posterior distribution, $p(z|y, \Theta) \propto p(z|\theta)p(y|z, \gamma)$, is often analytically intractable. In such cases, a common approach is to seek an approximation $q(z)$ constrained to a tractable class $\mathcal{Q}$ by variational inference (VI); that is, by minimising the KL-divergence to the true posterior distribution or, equivalently, maximising a variational free energy [6] $\mathcal{F}_{\text{VI}} = \langle \log p(y, z|\Theta) - \log q(z) \rangle_{q(z)}$:

$$q_{\text{VI}(\mathcal{Q})}(z|y, \Theta) = \underset{q \in \mathcal{Q}}{\text{argmin}} \, \mathbf{KL}[q(z)\|p(z|y, \Theta)] = \underset{q \in \mathcal{Q}}{\text{argmax}} \, \langle \log p(y, z|\Theta) - \log q(z) \rangle_{q(z)}. \quad (1)$$

In the variational autoencoder (VAE) architecture, the parametric optimisation of $q(z)$ implicit in (1) is amortised by a recognition network that takes as input the observed values $y$ and returns parameters of $q$. The parameters of this recognition network are then trained jointly with the generative model by stochastic optimisation of the free energy [1, 2].

In both standard and amortised VI, the approximate distribution $q$ is often constrained to factor over the latent variables, a so-called *mean-field* constraint. While tractable, such mean-field approximations may be too restrictive to capture the complexity of the true posterior [14, 15]. Many approaches have been proposed for improving the expressiveness of variational approximation [16–19]. Here we consider the structured VAE [SVAE; 7]. Unlike standard VAEs, the SVAE assumes the generative prior distribution to be specified by a structured probabilistic graphical model (PGM), $p(z|\theta) \propto \prod_{c \in C} \psi_c(z_c)$, where $\{\psi_c\}_{c \in C}$ correspond to $C$ clique potentials. In addition, the amortised inference network outputs recognition factors, $r(z|y, \phi)$, that approximate the generative likelihood function rather than the full variational posterior. These recognition factors are combined with the structured prior distribution to obtain the amortised variational posterior,

$$q_{\text{SVAE}}(z|y, \Theta, \phi) = \underset{q \propto r(z|y, \phi)p(z|\theta)}{\text{argmax}} \sum_{y \in \mathcal{Y}} \left\langle \log \left( \frac{p(z|\theta)p(y|z, \gamma)}{q(z)} \right) \right\rangle_{q(z)} \propto r(z|y, \phi^*)p(z|\theta), \quad (2)$$

where $\phi^*$ minimises the averaged KL over the data observations $\mathcal{Y}$. Johnson et al. [7] considered recognition factors that are local (singleton) evidence potentials, chosen to be conjugate to $p(z|\theta)$. Even in this case, the form of (2) allows the dependency structure established by the prior to be carried over to the variational posterior distribution. Further details of SVAE appear in Appendix A.

### 2.2 Gaussian Process Factor Analysis

Gaussian Process Factor Analysis (GPFA) is a model used in neural data analysis to infer the dynamic latent structure underlying high-dimensional population spike trains [11, 12]. Standard GPFA assumes

the following generative model:

$$\text{latent functions: } f^k(\cdot) \sim \mathcal{GP}\big(m_\theta^k(\cdot), \kappa_\theta^k(\cdot, \cdot)\big), \text{ for } k = 1, \dots, K,$$

$$\text{affine embeddings: } h_n(\cdot) = \sum_{k=1}^{K} c_{nk} f^k(\cdot) + d_n, \text{ for } n = 1, \dots, N, \tag{3}$$

$$\text{observations: } y_n(t) \sim p\big(y_n(t) | g(h_n(t))\big), \text{ for } t = 1, \dots, T,$$

where $m_\theta^k(\cdot)$ and $\kappa_\theta^k(\cdot, \cdot)$ parametrise the GP prior, $c_{jk}$ and $d_n$ define an affine mapping from the latent space to the observation space (or a transform of the observations for general likelihoods), and $g(\cdot)$ is a smooth scalar link function appropriate for the observation distribution. We take $m_\theta^k(\cdot) = 0$ unless stated otherwise. Correlations between observed neurons are captured by dependence on the common set of latents, while temporal correlations in the high-dimensional observations are modelled by the latent temporal correlations of the GPs. The PGM describing the GPFA generative model (with sparse variational approximation, see below) is shown in Fig 1a.

Although maximum-likelihood parameters for GPFA can be found using (variational) expectation-maximisation [20, 11] exact GP inference scales cubically in the number of observation times [21]. Sparse variational GP (svGP) inference based on auxilliary inducing points [22] reduces the time complexity of learning in GPFA [23], and also facilitates efficient extensions to (pointwise) non-conjugate likelihoods [12].

For $k = 1, \dots, K$, we introduce inducing values $\mathbf{u}_k$ representing the evaluations of the latent process $f^k$ at $M_k$ inducing locations, $\mathbf{z}_k$. For simplicity, we assume $M_1 = \dots = M_K = M$ unless stated otherwise. The GPFA generative model (3) can be augmented to include these auxiliary variables:

$$p(\mathbf{u}_k | \mathbf{z}_k) = \mathcal{N}\big(\mathbf{0}, \mathbf{K}_{\mathbf{z}_k \mathbf{z}_k}^k\big), \quad p(f^k(\cdot) | \mathbf{u}_k) = \mathcal{GP}(\mathfrak{F}_k(\cdot) \mathbf{u}_k, \kappa_\theta^k(\cdot, \cdot) - \mathfrak{F}_k(\cdot) \mathbf{K}_{\mathbf{z}_k \mathbf{z}_k}^k \mathfrak{F}_k^T(\cdot)), \tag{4}$$

where $\mathfrak{F}_k(\cdot) = \kappa_\theta^k(\cdot, \mathbf{z}_k)(\mathbf{K}_{\mathbf{z}_k \mathbf{z}_k}^k)^{-1}$ is the linear operator that maps $\mathbf{u}_k$ to $f^k(\cdot)$; $\kappa_\theta^k(\cdot, \mathbf{z}_k)$ is a vector-valued function with $\kappa_\theta^k(x; \mathbf{z}_k) = [\kappa_\theta^k(x, z_{k1}), \kappa_\theta^k(x, z_{k2}), \dots, \kappa_\theta^k(x, z_{kM_k})]$; and $\mathbf{K}_{\mathbf{z}_k \mathbf{z}_k}^k$ is the covariance matrix obtained by evaluation of the kernel function $\kappa_\theta^k(\cdot)$ at the inducing locations $\mathbf{z}_k$.

Introducing a variational distribution over the inducing points and the latent functions, and utilising the generative model (Fig 1a), the free energy of the sparse variational GPFA (svGPFA) has the form

$$\mathcal{F}(\Theta, \mathbf{C}, \mathbf{d}, \mathbf{Z}, \phi) = \left\langle \log \frac{\left(\prod_{t=1}^T p(\mathbf{y}_t | \mathbf{f}_t; \gamma, \mathbf{C}, \mathbf{d})\right) p(\mathbf{F} | \mathbf{U}; \theta) p(\mathbf{U} | \mathbf{Z}; \theta)}{q(\mathbf{F}, \mathbf{U}; \phi)} \right\rangle_{q(\mathbf{F}, \mathbf{U}; \phi)}. \tag{5}$$

where (with $[x_i]$ denoting an array obtained by iterating over the index $i$), $\mathbf{y}_t = [y_n(t)] \in \mathbb{R}^N$, $\mathbf{f}_t = [f^k(t)] \in \mathbb{R}^K$, $\mathbf{F} = [\mathbf{f}_t^T] \in \mathbb{R}^{T \times K}$, $\mathbf{C} = [c_{nk}] \in \mathbb{R}^{N \times K}$, $\mathbf{d} = [d_n] \in \mathbb{R}^N$, $\mathbf{U} = [\mathbf{u}_k]^T \in \mathbb{R}^{K \times M}$, and $\mathbf{Z} = [\mathbf{z}_k]^T \in \mathbb{R}^{K \times M}$. The svGP approach constrains $q(\mathbf{F}, \mathbf{U}; \phi)$ to the form $q(\mathbf{U}) p(\mathbf{F} | \mathbf{U}, \theta)$ with Gaussian $q(\mathbf{U})$ [22]. In the multi-GP case, the distribution is typically also taken to factorise over processes [23], $q(\mathbf{F}, \mathbf{U}) = \prod_{k=1}^K p(\mathbf{f}^k | \mathbf{u}_k) q(\mathbf{u}_k)$, with $\mathbf{f}^k = [f^k(1), \dots, f^k(T)] \in \mathbb{R}^T$, $q(\mathbf{u}_k) = \mathcal{N}(\mathbf{m}_k, \mathbf{S}_k)$. Under these constraints on $q$, the svGPFA free energy simplifies to

$$\mathcal{F}_{\text{svGPFA}} = \sum_t \langle \log p(\mathbf{y}_t | \mathbf{h}_t) \rangle_{q(\mathbf{h}_t)} - \sum_{k=1}^K \mathbf{KL}[q(\mathbf{u}_k) \| p(\mathbf{u}_k | \mathbf{z}_k)],$$

where $q(\mathbf{h}_t) = \int d\mathbf{U} \, p(\mathbf{h}_t | \mathbf{U}) q(\mathbf{U})$ is itself a GP with mean and kernel functions given by

$$m_n^h(t) = \sum_{k=1}^K c_{nk} \mathfrak{F}_k(t) \mathbf{m}_k + d_n \text{ and } \nu_n^h(t, t') = \sum_{k=1}^K c_{nk}^2 (\kappa_k(t, t') + \mathfrak{F}_k(t)(\mathbf{S}_k - \mathbf{K}_{\mathbf{z}_k \mathbf{z}_k}) \mathfrak{F}_k(t')^T).$$

Hence the computational cost for sparse variational GP inference reduces from $\mathcal{O}(T^3)$ down to $\mathcal{O}(M^3 + TM^2)$.

## 3 Structured Recognition and Explaining Away

### 3.1 Structured Recognition in VAEs

A general non-linear likelihood may induce additional latent dependency structure in the posterior beyond that of the prior (a simple illustrative example appears in Appendix C). Although the SVAE

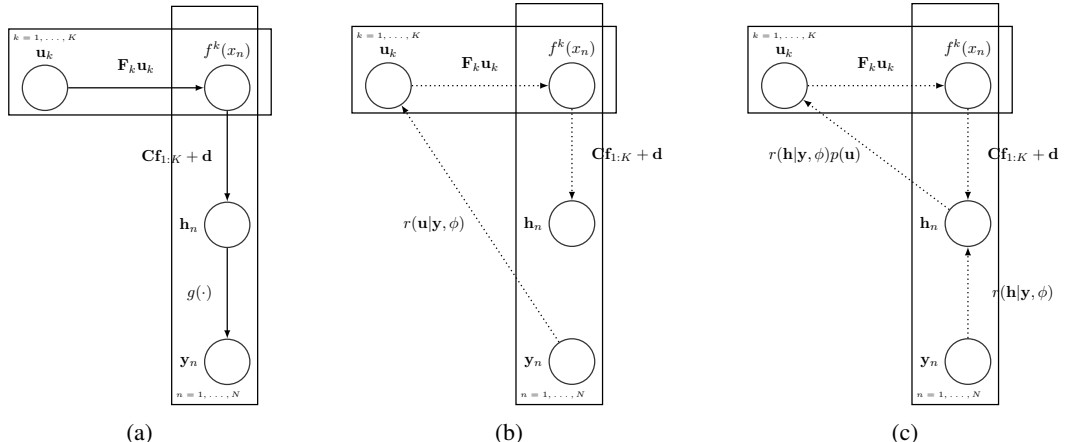

Figure 1: **Graphical models for sparse variational GPFA models**. **(a)** Generative model of GPFA model with sparse inducing point approximation; **(b)** Standard GPFA inference with sparse amortised variational approximation (note that $\mathbf{C} = \mathbf{I}$ and $\mathbf{d} = \mathbf{0}$ for SGP-VAE [10]); **(c)** Structured recognition potential enables full-covariance variational inference in SR-nlGPFA model.

approach developed by Johnson et al. [7] composed amortised inference with a structured prior distribution, the recognition models these authors discussed all contributed amortised potentials that factored over latent variables. Such an approach cannot accurately model posteriors with dependence structure that differs from that of the prior PGM.

Here we adopt a recognition network that outputs structured factor potentials over the latents, providing amortised estimates of the additional dependencies induced by "explaining away". We denote the resulting model the *Structured Recognition VAE* (SRVAE). The SRVAE variational approximation takes the form

$$q_{\text{SRVAE}}(z|y, \theta, \phi) \propto \prod_{c \in C} \psi_c(z_c; \theta) \prod_{c \in C_r} \xi_c(z_c|y; \phi), \tag{6}$$

where $C_r$ is the set of recognition factors and the $\xi_c$s the associated factor potentials. We assume that the $\xi_c$s are chosen to be conjugate to the prior factors unless otherwise stated. Hence the analytical (approximate, if necessary) form of $q_{\text{SRVAE}}$ can be computed with (variational) message passing. In the most general form, we could let $\xi(z|y; \phi)$ be a single joint factor potential over the complete set of latent variables.

The free energy objective takes the same form as the standard autoencoding free energy objective [1, 2], but with the structured amortised variational approximation, $q_{\text{SRVAE}}$, as the variational distribution.

$$\mathcal{F}_{\text{SRVAE}}(\Theta, \phi) = \langle \log p(z|\theta) + \log p(y|z, \gamma) - \log q_{\text{SRVAE}}(z|y, \phi, \theta) \rangle_{q_{\text{SRVAE}}(z|y, \phi, \theta)} \tag{7}$$

The training of the model follows standard VAE-style stochastic optimisation to update the parameters of the recognition and generative networks, as well as (optionally) the parameters of the prior PGM.

The following proposition supports the use of SRVAE framework (the proof appears in Appendix E).

**Proposition 3.1.** *The SRVAE objective function provides a tighter lower bound to the free energy than the SVAE objective function.*

$$\max_q \mathcal{F}_{VI(\mathcal{Q})}(\Theta, q) \geq \max_\phi \mathcal{F}_{SRVAE}(\Theta, \phi) \geq \max_\phi \mathcal{F}_{SVAE}(\Theta, \phi) \tag{8}$$

Below we describe two instantiations of the SRVAE framework, the first based on a latent Gaussian mixture model, and the second on a latent GPFA model, both with DGM outputs. Structured amortised inference facilitates scalable inference of the posterior latent distribution with full covariance structure, allowing more accurate learning than with factored recognition approaches. However, the SRVAE framework is more general-purposed, and can be combined with many different latent variable models. Further examples appear in Appendix E.

## 3.2 Structured Recognition with Latent Gaussian Mixture Model

The first experiments in Section 5 apply SRVAE to the setting of a latent Gaussian mixture model (GMM) [24] of the form (Figure 5a)

$$z_t|\pi \sim \text{Categorical}(z|\pi)$$
$$\mathbf{h}_t \sim \mathcal{N}(\mathbf{h}|\boldsymbol{\mu}^{(z_t)}, \boldsymbol{\Sigma}^{(z_t)}) \tag{9}$$
$$\mathbf{y}_t|\mathbf{h}_t \sim \mathcal{N}(\mathbf{y}|\boldsymbol{\mu}_{\text{NN}}(\mathbf{h}_t), \boldsymbol{\Sigma}_{\text{NN}}(\mathbf{h}_t)).$$

This model was studied by Johnson et al. [7], who used an SVAE with fully factorised recognition potentials and variational message passing[VMP; 25] to obtain the GMM variational posterior (details can be found in Appendix A). However, the neural-network defined conditional model induces posterior correlations between the latent variables beyond those imposed by the prior. Thus here we used an SRVAE model, with a full covariance Gaussian recognition potential on $\mathbf{h}$, combined with the prior factor using VMP.

## 3.3 Structured Recognition Variational Autoencoding Nonlinear GPFA

The GPFA model (3) incorporates a generalised-linear likelihood, with the link function $g(\cdot)$ acting separately on each affine embedding value $h_n$. We are now in a position to use the SRVAE framework to extend GPFA to include a DGM likelihood, greatly increasing the expressiveness of the generative model. Consider observations $\{(x_t, \mathbf{y}_t)\}_{t=1}^T$ and define $\mathbf{f}_t \in \mathbb{R}^K$ and $\mathbf{h}_t \in \mathbb{R}^N$ to be the corresponding vectors of latent process values and embeddings at $x_t$. [In neural applications, the input $x_t$ is usually taken to be identical to the timestamp $t$; see (3). Here we consider the more general case of arbitrary time-dependent inputs.] The generative model we consider retains the affine mapping from $\mathbf{f}_t$ to $\mathbf{h}_t$, but replaces the link function by a non-linear multivariate DGM: $\mathbf{y}_t \sim p(\mathbf{y}|g(\mathbf{h}_t, \gamma))$. Although the generative affine mapping could, in principle, be subsumed within the general deep network, the embeddings $\mathbf{h}_t$ will play a valuable role in parametrising structured recognition.

For amortised svGP inference, our goal is to define a variational distribution $q(\mathbf{U}|\mathbf{Y}, \phi)$. Ashman et al. [10] have considered a similar latent GP model called the SGP-VAE. There, the variational distribution over the latents was found by combining the prior GP distribution with an amortised approximation of the likelihood function factored over latent processes [22]:

$$q(\mathbf{F}, \mathbf{U}) = \prod_k p(\mathbf{f}^k|\mathbf{u}_k)q(\mathbf{u}_k), \text{ where } q(\mathbf{u}_k) \propto p(\mathbf{u}_k) \prod_t r(\mathbf{u}_k|\mathbf{y}_t, x_t, \mathbf{z}_k), \tag{10}$$

The graphical model for SGP-VAE inference corresponds to that of Figure 1b, with $\mathbf{C} = \mathbf{I}$ and $\mathbf{d} = \mathbf{0}$. While this approach captures induced correlation amongst the inducing points, it fails to capture correlations *between* latent processes that arise in the posterior through "explaining away", potentially leading to sub-optimal inference and learning.

Our solution is to recast the output of the amortised inference to the likelihood of $\mathbf{h}$ rather than $\mathbf{U}$, and propose the following structured variational distribution.

$$q(\mathbf{F}, \mathbf{U}) = \left[\prod_k p(\mathbf{f}^k|\mathbf{u}_k)\right] q(\mathbf{U}), \text{ where } q(\mathbf{U}) \propto \int d\mathbf{H}\, p(\mathbf{U})p(\mathbf{H}|\mathbf{U}) \prod_t r(\mathbf{h}_t|\mathbf{y}_t) \tag{11}$$

and $r(\mathbf{h}_t|\mathbf{y}_t) \propto \mathcal{N}(\mathbf{h}_t|\mathbf{m}(\mathbf{y}_t;\phi), \boldsymbol{\Psi}(\mathbf{y}_t;\phi)) = \mathcal{N}(\boldsymbol{\mu}_t^h, \boldsymbol{\Psi}_t^h)$, with $\boldsymbol{\Psi}_t^h$ diagonal, and $\mathbf{H} = [\mathbf{h}(x_1), \ldots, \mathbf{h}(x_T)]$. Given the linear-Gaussian relationship between $\mathbf{U}$ and $\mathbf{h}$ (Fig 1a), the integral in (11) can be computed in closed form. Furthermore, even though the recognition potential on $\mathbf{h}$, $r(\mathbf{h}|\mathbf{y}_t)$, is assumed to be fully factorised over the dimensions of each $\mathbf{h}_t$, the variational distribution on $\mathbf{U}$ includes coupling between latent processes induced by the affine mixing coefficients. Hence $q(\mathbf{U})$ captures the correlations both between the latent processes and through time (through combination with the GP prior under the structured autoencoding formulation).

Specifically, given the general expression of the GPFA generative model and the sparse approximation with inducing points, the linear Gaussian relationship between $\mathbf{h}$ and $\mathbf{U}$ at any $x$ is given by

$$p(\mathbf{h}|\mathbf{U}, x) = \mathcal{N}\left(\mathbf{C}\mathfrak{F}(x)\mathbf{U} + \mathbf{d}, \mathbf{C}(\mathbf{K}_x - \mathfrak{F}(x)\mathbf{K}_\mathbf{U}\mathfrak{F}(x)^T)\mathbf{C}^T\right), \tag{12}$$

where

$$\mathfrak{F}(x) = \begin{bmatrix} \mathfrak{F}_1(x) & & \\ & \ddots & \\ & & \mathfrak{F}_K(x) \end{bmatrix}, \quad \mathbf{K_U} = \begin{bmatrix} \mathbf{K_{z_1 z_1}} & & \\ & \ddots & \\ & & \mathbf{K_{z_K z_K}} \end{bmatrix}$$

with $\mathbf{K}_x = \mathrm{diag}[\kappa_\theta^k(x,x)]$. Combining this result with the fully factorised recognition potential on $\mathbf{h}$ at $x_t$ (11), we obtain the structured variational distribution on $\mathbf{U}$:

$$q(\mathbf{U}) = \mathcal{N}(\mathbf{m_U}, \mathbf{S_U}) \propto p(\mathbf{U}) \prod_t \mathcal{N}(\mathbf{C}\mathfrak{F}(x_t)\mathbf{U}|\boldsymbol{\mu}_t^h, \boldsymbol{\Psi}_t^h), \text{ with} \tag{13}$$

$$\mathbf{S_U^{-1}} = \mathbf{K_U^{-1}} + \sum_t \mathfrak{F}(x_t)^T \mathbf{C}^T (\boldsymbol{\Psi}_t^h)^{-1} \mathbf{C}\mathfrak{F}(x_t), \quad \mathbf{m_U} = \mathbf{S_U}\left(\sum_t \mathfrak{F}(x_t)^T \mathbf{C}^T (\boldsymbol{\Psi}_t^h)^{-1}(\boldsymbol{\mu}_t^h - \mathbf{d})\right).$$

This variational posterior on $\mathbf{U}$ leads to a corresponding posterior on $\mathbf{h}$ at each $x$:

$$q(\mathbf{h}(x)) = \int d\mathbf{U}\, p(\mathbf{h}|\mathbf{U})q(\mathbf{U}) = \mathcal{N}\big(\mathbf{C}\mathfrak{F}(x)\mathbf{m_U} + \mathbf{d}, \mathbf{C}(\mathbf{K}_n + \mathfrak{F}(x)(\mathbf{S_U} - \mathbf{K_U})\mathfrak{F}(x)^T)\mathbf{C}^T)\big).$$

and we write $q(\mathbf{h}(x_t)) = \mathcal{N}\big(\mathbf{m}_t^h, \mathbf{S}_t^h\big)$. Then the complete (reparametrised) Monte Carlo estimate of the free energy objective given a mini-batch of data, $\{(x_b, \mathbf{y}_b)\}_{b=1}^B$, takes the following expression.

$$\mathcal{F}(\theta, \gamma, \phi, \mathbf{C}, \mathbf{d}, \mathbf{Z}) = \sum_{b=1}^B \frac{1}{S} \sum_{s=1}^S \log p(\mathbf{y}_b | \mathbf{m}_b^h + \mathbf{L}_b^h \epsilon_s) - \mathbf{KL}[q(\mathbf{U})\|p(\mathbf{U})] \tag{13}$$

where $\epsilon_s \sim \mathcal{N}(\mathbf{0}, \mathbf{I})$, $\theta$ is the set of kernel parameters, $\mathbf{L}_b^h$ is the lower-triangular Cholesky component of $\mathbf{S}_b^h$ such that $\mathbf{S}_b^h = \mathbf{L}_b^h(\mathbf{L}_b^h)^T$.

We note that we have retained the scalability of amortised inference by choosing an amortised diagonal-Gaussian potential on $\mathbf{h}$. However, the linear-Gaussian relationship between the inducing points $\mathbf{U}$ and $\mathbf{h}(x)$ (12) leads to a full-covariance variational Gaussian approximation for $\mathbf{U}$, allowing amortised inference to capture the observation-induced posterior correlations between latent processes (commonly referred to as the "explaining away" effect). Thus, the retention of the affine GPFA mapping introduces a key extension to SGP-VAE. We will refer to the new model as *structured recognition non-linear GPFA* (SR-nlGPFA). The generative model corresponds to that of svGPFA (Figure 1a) but with the nonlinearity $g(\cdot)$ generalised to a flexible form modelled by a neural network. The complete inference procedure for SR-nlGPFA is graphically illustrated in Figure 1c.

The implicit computation of the variational distribution over $\mathbf{U}$ given the recognition potentials on $\mathbf{h}$ also makes it possible to carry out svGP inference with changed inducing locations (13). This is particularly useful in situations that require inference over test datasets of different durations to those seen in training. In particular, it may be possible to learn an amortised inference network using short sub-sequences drawn from a longer dataset, and then infer a posterior over latent GPs for the complete data sequence efficiently by optimising the placement of inducing points along its full length. See further details on the free-form svGP inference step in Appendix F.2.

## 4 Related work

**Structured Deep Generative Models.** A number of studies have considered latent graphical structure within the DGM framework [26, 27, 7, 28–30]. Our work builds on and generalises the prominent SVAE proposal of Johnson et al. [7]. Both SVAE and SRVAE combine the structure of a prior PGM with the flexibility of neural-network-based recognition. The difference lies in the form of the recognition potentials. These were taken to extend over single latent variables in the earlier study. Here, we consider joint potentials that capture dependence induced by the generative likelihood. In a closely related study, Lin et al. [28] proposed a structured inference network, which approximates the variational distribution as the combination of the recognition potential and a separate structured latent distribution independent of the prior, the independent structured latent distribution can have non-conjugate factors to improve the expressiveness of the model.

**svGPFA and Extensions.** Since its introduction as a model to identify linear low-dimensional struction in neural population data [11], GPFA has been extended to incorporate non-linear link

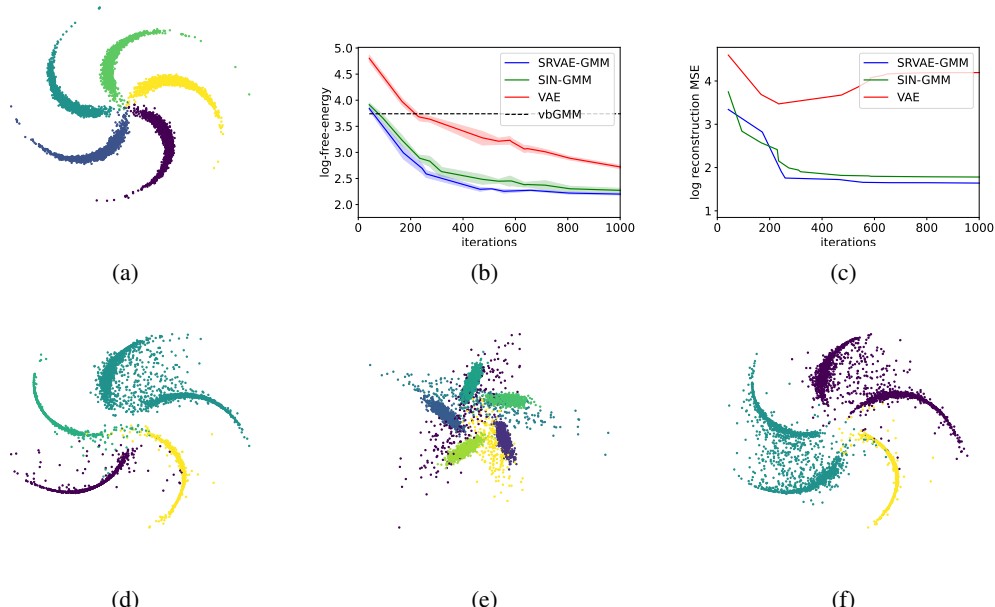

(a)                            (b)                            (c)

(d)                            (e)                            (f)

Figure 2: **Empirical Evaluations on Pinwheel Dataset.** (a) Sampled data from the pinwheel dataset; Training curves of the (b) variational free energy objective and (c) testing reconstruction MSE through out training (all in log-scale); Sampled data given trained model of (d) SRVAE-GMM; (e) variational GMM; (f) SIN-GMM [28]. All evaluations are based on the averages over 5 random seeds.

functions and non-conjugate (notably Poisson count or point-process) noise models, and combined with sparse variational inference for efficiency [23, 31, 12, 32]. Related work has considered GP latents in the context of DGMs, particularly using variational autoencoding [9, 33, 10]. In a sense, SR-nlGPFA combines both approaches. It incorporates multiple latent GPs (as in GPFA and some DGM GP models) with an affine mapping that feeds into a nonlinear DGM. However, this structure means that the affine map contributes little to the generative process. Instead it provides a target for amortised inference, which combines with sparse variational GP inference to yield a full structured posterior on inducing point values. As such, the affine map may arguably be better seen as an element of the structured recognition model.

## 5 Results

We evaluated SRVAE methods for both latent mixtures, and nlGPFA, comparing to relevant baselines on synthetic and real datasets. Empirical results on the instantiations of the SRVAE framework with other latent variable models can be found in Appendix F[1].

### 5.1 Experiments with SRVAE-GMM

To assess the empirical performance of structured amortisation in a standard latent-variable model, we evaluated SRVAE-GMM on the pinwheel dataset of [7] (Figure 2a). We compare SRVAE-GMM with the structured inference network [SIN; 28], which is a more flexible instantiation of the SVAE framework. From Figure 2, we observe that SRVAE-GMM outperforms SIN-GMM in terms of the training variational free energy, reconstruction mean-squared error, and generation fidelity. The models are identical apart from the recognition stage, providing support for the idea that by capturing the "explaining away" posterior factors, structured recognition leads to a more accurate learnt model.

---

[1]Python implementation can be found at `https://github.com/gatsby-sahani/structured-recognition-neurips2022`

|  | Synthetic data | | EEG | | Population spiking |
| --- | --- | --- | --- | --- | --- |
|  | SMSE | NLL | SMSE | NLL | SMSE |
| SR-nlGPFA | **0.31 ± 0.02** | **1.69 ± 0.04** | **0.27 ± 0.05** | **1.81 ± 0.21** | **0.47 ± 0.18** |
| SGP-VAE [10] | 0.36 ± 0.05 | 1.76 ± 0.07 | 0.35 ± 0.05 | 2.17 ± 0.17 | 0.55 ± 0.19 |
| Vanilla-VAE [1, 2] | 1.05 ± 0.02 | 13.60 ± 3.58 | 0.57 ± 0.09 | 3.45 ± 0.87 | 3.19 ± 1.98 |
| SVAE-LDS [7] | 0.933 ± 0.02 | 9.37 ± 1.94 | 3.04 ± 0.38 | 11.71 ± 2.66 | 2.52 ± 1.31 |

Table 1: Quantitative comparison of performance between SR-nlGPFA and baseline models on synthetic dataset, EEG dataset and population spiking dataset (Section 5.2.2) with respect to standarised mean squared error (SMSE) and negative log-likelihood (NLL). Averages over 5 random seeds.

## 5.2 Experiments with SR-nlGPFA

### 5.2.1 Synthetic and EEG Dataset

We compared SR-nlGPFA to methods that do not capture the inter-latent posterior correlation, using synthetic data and a small-scale EEG dataset used previously [34, 10]. Unless stated otherwise, we employed an exponentiated quadratic kernel, $\kappa(x, x') = \lambda \exp(-\frac{||x-x'||^2}{\tau^2})$, where $\lambda$ and $\tau$ are the marginal variance and length-scale parameters, respectively.

**Baselines** Our main baseline model is SGP-VAE [10] (described in Sections 3.3 and 4). We also compared to a "vanilla" VAE with fully factorised Gaussian variational distribution [1, 2], and SVAE with a linear dynamical system (SVAE-LDS) latent prior [7]. All models were implemented with the same recognition and generative network architectures (see Appendix G for implementation details).

**Synthetic Dataset** We generated data from the GPFA generative model (Eq. 3), with $g(\cdot) = \sigma(\Phi(\cdot))$, where $\Phi(\cdot)$ represents the functional mapping through a fixed, randomly-initialised 2-layer MLP with ReLU hidden non-linearity, and $\sigma(\cdot)$ is the sigmoid function.

**EEG Dataset** We follow the experimental procedure described by Requeima et al. [34], and consider an EEG measurement dataset spanning 1 second at 256 Hz sample frequency, taken during image viewing [35]. Each datapoint consists of the voltage readings from 7 electrodes positioned on the participant's scalp. Here we report results when all data are observed, contrary to the settings in [34, 10] (we include results and further discussion on partial observability in Appendix F.2).

SR-nlGPFA achieved lower standardised mean squared error (SMSE) and lower test negative log-likelihood (NLL) than the alternative methods on both data sets (Table 1), providing evidence for the benefits of structured recognition. These gains come despite the computational complexity of SR-nlGPFA being of the same order as SGP-VAE (see appendix F.2).

### 5.2.2 Population Neuronal Firing Data

The firing of place cells in Hippocampal area CA1 and grid cells in the medial Entorhinal Cortex (mEC) is known to be modulated by the animal's location [36, 37], speed and direction of locomotion [38–40], with the mapping between these behavioural covariates and neural activity expressed in non-linear mixed tuning curves. The behavioural signals are continuous and often mutually dependent, and so create temporal and spatial structure in the time series of population activity. We asked whether SR-nlGPFA and related methods would be able to identify and extract this structure without supervision; that is, without direct access to the behavioural covariates.

We used single-cell spiking data from neurons in the hippocampal CA1 and mEC regions of rats recorded during exploration of a Z-shaped track, as reported by Ólafsdóttir et al. [41]. The data comprised 28 experimental sessions, each spanning 10 minutes. Example neural firing patterns of the population are shown in Figure 11 of the Appendix.

For SR-nlGPFA and SGP-VAE, we adopted the GPFA generative model Eq. 3 with DGM non-linearity and Poisson observation likelihood.

$$p(\mathbf{y}(x)|\mathbf{h}(x)) = \prod_{n=1}^{N} \text{Poisson}(y_n(x)|g(\mathbf{h}(x))_n) \tag{14}$$

where $x$ are discrete times, $N$ is the number of recorded neurons, and $g(\cdot)$ represents the neural-network mapping from the GPFA features, $\mathbf{h}$, to the rate of Poisson-distributed firing counts for each neuron over contiguous 100 ms bins. SR-nlGPFA obtained a lower SMSE of prediction than SGP-VAE and other methods tested (Table 1).

Computational constraints meant that the SR-nlGPFA model was trained using short batches of data and 64 inducing points per batch. Latent trajectory estimates derived from such batches will not necessarily be continuous at the boundaries between them. Thus, once the model was fit, we performed svGP inference over complete sessions with increased numbers of inducing points (see Section 3.3 and Appendix F.3.2).

To relate these recovered latent time-series to behavioural covariates, we performed two-dimensional Canonical Correlation Analysis [CCA; 42]. We present results for one session here, emphasising qualitative effects. Further qualitative and quantitative results across sessions can be found in Appendix F.3.2.

Figure 3a shows a heatmap of the correlation coefficients between the canonical correlates of the posterior means ($CCX\{1, 2\}$) and the individual behavioural covariates (distance from one end of the track, speed, direction of travel, head direction, and 'unfolded' position along a full lap of the track). Many correlations are high, indicating the low-dimensional manifold parametrised by the conjunctive set of behavioural correlates can be accurately captured by the posterior latent variables learned with SR-nlGPFA solely from neural spikes. Please refer to Appendix Figure 13 for numerical values of the correlations (and for other sessions).

To see whether the learned latent structure contained decodable information about behavioural covariates, we extracted the direction-modulated neurons predicted by the trained model (details in Appendix F.3.2). Figure 3b compares the direction modulation of the model-predicted neurons against that of the rest of the neurons, where direction modulation is defined as the correlation between raw single-cell spike counts and the direction values. Neurons predicted by the model to be direction modulated exhibited significantly greater direction modulation than the other neurons (p-value=$5.56 \times 10^{-4}$) [43]. Figure 3c shows the firing profile for travel in each direction for the neuron with the strongest predicted direction modulation. The pattern of firing fields exhibits clear directional dependence. Similar comparisons hold for spatial and speed modulation, indicating that SR-nlGPFA is able, in a purely unsupervised fashion, to learn a latent space that contains linearly decodable information about the behavioural covariates associated with individual neuronal firing, even for neurons whose activity exhibits conjunctive coding.

Figure 3d shows the two canonical correlates obtained from the latent trajectories as a function of the animal's spatial location. These reflect both location and direction of movement (colours), indicating the latent dimensions learned by SR-nlGPFA disentangle direction from spatial location in a simple linear projection, despite the conjunctive coding mechanism exhibited in the CA1 neurons. A similar plot derived from SGP-VAE (Figure 14b), shows less clear disentanglement.

Figure 3e shows the temporal evolution of the latent CCs for SR-nlGPFA and SGP-VAE, as well as the spatial location of the rat. Both $CCX1$ and $CCX2$ of SR-nlGPFA consistently track the dynamics of the distance (with perfect phase alignment and half-cycle offset, respectively), whereas neither $CCX1$ or $CCX2$ of SGP-VAE could accurately reflect the distance. Moreover, given the additional svGP inference step over the entire trajectory (with increased number of inducing points), latent signals of SR-nlGPFA exhibit significantly greater smoothness than that of SGP-VAE.

## 6 Discussion

We have developed a framework of structured recognition for variational autoencoders, which can be viewed as a strict generalisation of the SVAE [7], capturing posterior correlations induced by the "explaining away" effect. Correspondingly, the SRVAE framework yields a tighter bound on the log-likelihood than the SVAE free energy objective.

Applying structured recognition and non-linear variational autoencoding to GPFA, we introduce SR-nlGPFA, facilitating scalable free-form variational Gaussian approximation. SR-nlGPFA supports a fully non-linear, non-conjugate extension to standard GPFA [11, 23, 12], which is particularly well-suited to settings where posterior inference must be scaled to additional data. We show that SR-nlGPFA outperforms the baseline on all presented tasks, both quantitatively and qualitatively.

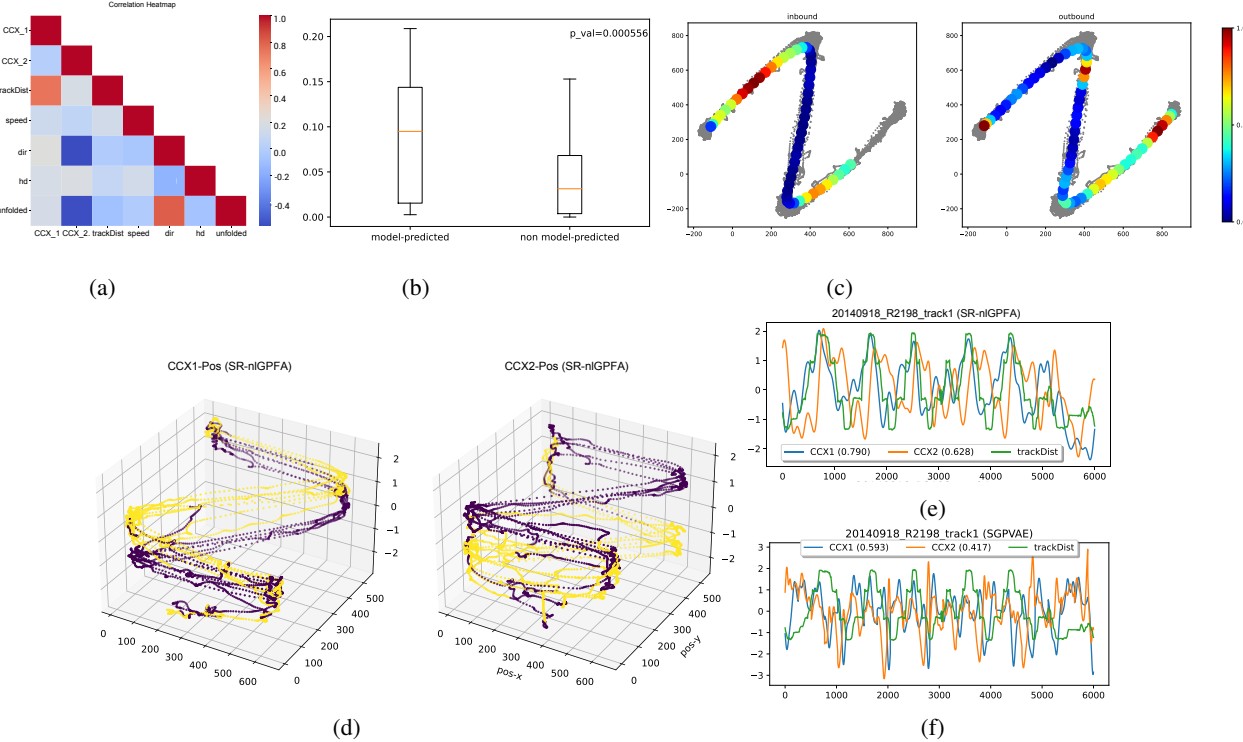

Figure 3: **Analysis of SR-nlGPFA posterior latent structure**. **(a)** Heatmap of the correlation between the CCs and the behavioural covariates for the selected session; **(b)** Directional-modulation comparison between the model-predicted direction-modulated neurons against other neurons; **(c)** Exemplary firing fields of the model-predicted direction-modulated neuron during inbound v.s. outbound movements; **(d)** Plot of $CCX\{1, 2\}$ of SR-nlGPFA posterior latents against spatial location, color indicates direction of movement (yellow: inbound, magenta: outbound); **(e-f)** Temporal evolution of (standardised values of) $CCX\{1, 2\}$ and spatial location for SR-nlGPFA and SGP-VAE.

Studying population spiking data of hippocampal neurons, we show that SR-nlGPFA is able to identify latent signals that exhibit strong correlation with the behavioural covariates that are well-known to be conjunctively encoded by the recorded neurons, but without direct access to these covariates. While here we focused on temporal correlation exhibited in time-series data, we note that the sparse amortised variational approximation presented here can also be straightforwardly applied to other GP-latent models, such as GP-LVM [44, 45]. While we have considered spiking data alone, these could be substituted by or combined with EEG, calcium imaging, or indeed behavioural data recorded during the same session. Such joint analysis of latent structures given different recordings within the same session could potentially facilitate new findings with respect to, e.g., replay, phase coding, etc.

## 7   Acknowledgement

We thank the anonymous reviewers for helpful comments and discussions, and Zilong Ji and Dan Bush for helpful discussions regarding the experiments with the neural data. This work is funded by the UKRI, DeepMind, the Gatsby Charitable Foundation, the Simons Foundation and the Wellcome Trust.

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
