# OpenReview forum: "Structured Recognition for Generative Models with Explaining Away"
_NeurIPS.cc/2022/Conference — NeurIPS 2022 Accept_

### Official Review · Reviewer_9rkF · 2022-07-03

**Rating:** 6
**Confidence:** 3
**Soundness:** 3 good
**Presentation:** 2 fair
**Contribution:** 2 fair

**Summary:**

Update after author rebuttal:
thanks for your changes, I believe that the restructuring makes the paper clearer and easier to read. I have now increased my score to a weak accept.
I still believe that the idea of extending the SVAE to structured factor potentials is quite obvious, and that the strongest contribution of this work is the AEA-svGPFA model. Similarly to reviewer D7Ke however, I must admit I am not familiar enough with the applications of the AEA-svGPFA model and its related literature to properly assess the impact this work can have in the Neurips community if accepted. I will discuss this with the other reviewers and the area chair.


=======================

This paper introduces an extension to the Structured VAE framework that considers structured factor potential instead of fully factorized ones. This allows the model to better capture observation-induced posterior dependencies.

The framework is applied 1) to a tree structured model 2) to a GP factor analysis model for neural spiking time series, and outperforms competing methods in a number of synthetic and real-world experiment.

**Questions:**

See comments under "weaknesses" above.

**Limitations:**

Yes

**Strengths And Weaknesses:**

*STRENGTHS*

1. The paper proposes an interesting an novel application of the SVAE framework to the svGPFA model, which allows to better model posterior dependencies among variables while keeping the model computationally efficient
2. The AEA-svGPFA outperforms competing methods on neuronal firing data
3. The authors perform in depth discussions that show the impact of better modelling posterior correlations with the proposed model







*WEAKNESSES*

My main concerns for this paper, are the novelty of the idea and the clarity of which one is the main contribution.

I consider the proposed framework that extends the SVAE to structured factor potential as obvious, and as such with very limited novelty (same for proposition 3.1).
Even after reading the SVAE paper alone, I find it natural that for posterior distributions with more complex dependencies among variables one can use factors over multiple variable instead of a mean field approach.
This was also noted by the authors of the SVAE paper as a possible extension of their model, in section 3.3. of "Structured VAEs: Composing Probabilistic Graphical Models
and Variational Autoencoders" by Johnson et al, 2016 (notice that this is a different version of the paper from the one you cited).



As stated above, for me the main contribution and novelty of the paper is the AEA-svGPFA model, and the paper should make this clear.
I would make the AEA-svGPFA model the main focus of the paper, presenting it as an application of the standard SVAE framework to the svGPFA model.
Doing this, you can
1. Move the tree-structured latent PGM and the synthetic bar experiments to the appendix, since they would contribute very little to the main storyline
2. Reduce the SVAE-related discussions
3. Add to the main text more important details on the AEA-svGPFA that are now in the appendix, which would make the paper easier to read



Minor comments:
* Typo in line 260: is->if?
* In line 269 you mention Figure 14 that does not exist
* Figure 4 subplots are way too small to be read on a printed version of the paper

---

> ### Author Response · Authors · 2022-08-02
> **Thank you for your review, below we address each of your questions/comments.**
>
> We wish to thank the reviewer for the thoughtful comments and suggestions. Below we address each of the mentioned questions/concerns.
>
> - Q1: Novelty of structured amortised inference.
>
> We thank the reviewer for pointing out the alternative version of the SVAE paper. We are aware of the fact that SVAE framework could potentially be implemented with structured recognition potential, however this was not explicitly discussed and instantiated in both the original SVAE paper and subsequent papers (e.g., Lin, et al. 2018). Here, with the Tree-structured amortised inference, we concretely instantiate the structured recognition potential, and evaluate the superiority over mean-field recognition potential with structured prior, both theoretically and empirically. We have now provided an additional instantiation of the AEA framework with the latent GMM prior and empirically tested on the pinwheel dataset (Appendix A.2 and E.2 of the updated manuscript). Similar to the experiment on the bar dataset, we observe improvement in terms of free energy, reconstruction MSE and generation quality. We agree that implicitly in the original SVAE paper, structured amortised inference is possible, and here we believe our contribution is to concretely instantiate such idea and prove the advantages of using structured recognition potential in addition to structure prior distribution.
>
> - Q2: Rearrangement of the paper.
>
> We thank the reviewer for raising valuable editing suggestions for improving the paper. We have made changes to the manuscript accordingly, and the changes made are summarised in the general reply. Any further comments/suggestions are welcome.
>
> - Q3: Minor comments.
>
> -- **Typos.** We thank the reviewer for pointing out the typo, which we have corrected.
>
> -- **Figure 14 missing.** We thank the reviewer for pointing out the missing Figure 14, you may find the exemplary neuron firing pattern in Figure 11 of the updated manuscript.
>
> -- **Figure 4 too small to read.** We thank the reviewer for the suggestion. We have now increase the size of subplots in Figure 2 of the updated manuscript. We hope the updated figures are more readable on printed papers.
>
> References:
>
> [1] Lin, W., Hubacher, N. and Khan, M.E., 2018. Variational message passing with structured inference networks. arXiv preprint arXiv:1803.05589.

---

### Official Review · Reviewer_D7Ke · 2022-07-10

**Rating:** 6
**Confidence:** 3
**Soundness:** 3 good
**Presentation:** 3 good
**Contribution:** 2 fair

**Summary:**

This paper proposes to extend structured variational inference to two settings to enable the modeling intra-latent variable dependencies. The first setting is a variational auto-encoder where the variational distribution and the approximated likelihood function is defined on a set of predefined factors (in this paper, a tree). The second setting is a structured version of svGPFA for which one can compute the full covariance matrix. Experiments are performed on a series of synthetic and real data.

**Questions:**

SGPVAE is the only baseline for the AEA-svGPFA, yet SGPVAE itself is not yet published on a major venue. Would this be a valid enough comparison? Have you considered other (possibly simpler) baselines, for example, a vanilla VAE or an SVAE with a linear dynamic system latent at least?

**Limitations:**

When the order of dependency increase, tractability would become an issue. For example, on the Tree-structured VAE case, how would you discuss if the graph is loopy (e.g., add an edge from one leaf to another) or if the graph is high order dependency (e.g, if a leaf node also depend on its grand father)?

**Strengths And Weaknesses:**

Strengths:

- The proposed method is able to model more complicated dependency beyond VAEs with mean-field inference model and the original svGPFA
- In the GPFA setting, the proposed method has the same asymptotic time complexity as the baseline SGPVAE
- The experiments and visualization on the Population spike dataset seems to be valid

Weaknesses

- On the structured VAE thread
    - Generally, I am not so convinced on the structured VAE side of the model, as there exist other works that also consider tree-structured or higher-order dependency over the latent variables, and there exist further issues about higher-order dependency.
    - About existing work considering intra-latent dependencies, for example, [1] introduces structured latent variables over permutations and trees, and [2] considers the joint modeling of continuous and tree-structured latent variables. Both works have more detailed discussions (about aspects like modeling and gradient estimation) and are applied to more complicated structures (Math equations and human language syntax trees) than the Bar dataset used in this work.
    - The tighter variational bound in equation 5 may also not be view as significant (in the context of 2022), as there are works with exact likelihood [3], also works discussing whether tighter bound are necessarily better[4].
    - Given the existing literature, the tree-structured VAE on the Bar dataset seems not very significant.
- On the AEA-svGPFA thread
    - I am more convinced on this extensions in terms of both model sophistication (to facilitate the modeling of full covariance) and the empirical evaluation (especially the visualized dynamics).
    - Yet I have to admit that I am not very familiar with the related svGPFA literature, so I would not be very confident to say if the contribution on this thread is significant enough for acceptance. I am open to discussions from the authors about why this work is significant enough from the GPFA perspective (and increase my scores accordingly if convinced).

Additional minor comments

- The fonts of Figure 1 and 2 are two small to recognize without zooming
- In equation 5, the objective for SVAE is not defined

References

[1] Paulus et. al. NeurIPS 2020. Gradient Estimation with Stochastic Softmax Tricks.

[2] Kim et. al. ACL 2019. Compound Probabilistic Context-Free Grammars for Grammar Induction.

[3] Lou et. al. ICLR 2020. SUMO: Unbiased Estimation of Log Marginal Probability for Latent Variable Models.

[4] Rainforth. et. al. ICML 2018. Tighter Variational Bounds are Not Necessarily Better.

---

> ### Author Response · Authors · 2022-08-02
> **Thank you for your review, below we address each of your questions/comments.**
>
> We wish to thank the reviewer for the thoughtful comments and suggestions. Below we address each of the mentioned questions/concerns.
>
> - Q1: Significance of AEA with Tree-structured SVAE on the Bar dataset.
>
> We propose the AEA implementation on the Tree-structured SVAE (referred to as the TreeSVAE) as a toy example for illustrating the positive effects brought by the AEA framework. Namely, on the simple Bar dataset, we have firstly showed that the simple generative model yield non-trivial additional posterior dependency structure (Appendix B in the updated manuscript), and then showed that with the AEA framework, TreeSVAE outperforms standard SVAE and VAE in terms of variational free energy, reconstruction MSE on testing dataset, and generation quality (please see Figure 6, 8, and Table 2 in the updated manuscript). In Appendix A.2 and E.2 of the updated manuscript, we have provided an additional implementation of AEA framework with the latent Gaussian mixture model prior and tested on the pinwheel dataset (used in, e.g., Johnson, et al. 2016, Lin, et al. 2018), where we show that again, the AEA framework yields improvement in terms of free energy, reconstruction MSE and generation quality.
>
> We wish to thank the reviewer for referring us to the relevant literature, which we have now cited and provided accompanying discussions.
>
> - Q2: Tighter ELBO is not necessarily better.
>
> We thank the reviewer for pointing out the tighter ELBO does not in general yield better performance, both empirically and theoretically. However, tighter ELBO does indicate smaller distance (KL-divergence) between the variational approximation and the true posterior distribution. A major goal of the paper is variational inference, where our claim is that the additional posterior dependencies introduced by "explaining away" can be captured with the AEA framework, hence leading to more accurate posterior inference. Indeed as demonstrated in the AEA-svGPFA model, we see that the AEA-svGPFA posterior latent processes accurately captures different aspects of the underlying behavioural covariates that are suggested to influence the CA1 neurons firing (O'Keefe and Nadel, 1978), whereas our main baseline, SGP-VAE, fails to do so (see comparison in Figure 2 e-f and Figure 13 in the updated manuscript). We provide further discussion in Appendix D (after the proof ends) in the updated manuscript.
>
> - Q3: Significance of AEA-svGPFA.
>
> Previous literature on GPFA models (Yu, et al. 2008, Adam, et al. 2016, Duncker and Sahani, 2018, Keeley, et al. 2020, Jensen, et al. 2021) have predominantly focused on the standard GPFA generative model (Eq. 3 from the updated manuscript), where there exists a one-to-one mapping from the neural space (h) to the neuron firing (y), where $g(\cdot)$ is some smooth pointwise mapping. Here we generalise the GPFA generative model such that the mapping from the neural space $h$ to the neuron firings $y$ is some general non-linearity. Such design choice greatly expands the expressiveness of the generative model, but at the same time introduces non-conjugacy. Previous works have addressed \textit{pointwise} non-linearity with approximate inference (e.g., polynomial approximation in Keeley, et al. 2020, quadrature approximation in Duncker and Sahani, 2018). Here we utilise a recognition network to perform amortised variational approximation in a SVAE-style inference. To the best of our knowledge, AEA-svGPFA is the first model that combine amortised inference with sparse variational approximations for GPFA models.
>
> One closely related work, SGP-VAE (Ashman, et al. 2020), despite having identical generative model in terms of functional form (by setting C=I and d=0 in Eq. 3), is driven by a different motivation that incorporating GP prior structure into posterior latent GPs, which does not utilise the nice linear-Gaussian relationship between the inducing points and the neural features (h), hence cannot capture posterior correlations between different latent processes.
>
> Moreover, in SGP-VAE, since the recognition network directly outputs a recognition potential on the inducing points, it is not able to modify the inducing points for additional svGP inference steps. AEA-svGPFA is able to resolve such issue since the recognition potential is on the neural space (h), and we are free infer arbitrary variational distribution on the inducing points given the linear-Gaussian relationship between the inducing points and neural space. This is particularly important in the amortised inference setting, since given the mini-batch nature of the stochastic optimisation, the number of inducing points is relatively small comparing to the number of observations, hence posterior inference over the entire trajectory will exhibit the temporal chunking effect with a small number of inducing points. Please see Section 3.2 and Section 5.2 and Figure 2 e-f for further discussion on the temporal chunking effect.
>
> (Continued below...)

---

> > ### Author Response · Authors · 2022-08-02
> > **Thank you for your review, below we address each of your questions/comments (continued).**
> >
> > In summary, AEA-svGPFA supports a fully non-linear, non-conjugate extension to standard GPFA (Yu, et al. 2008, Adam, et al. 2016, Duncker and Sahani, 2018), and is the first model that is able to perform variational inference with full-covariance variational Gaussian approximation that captures intra-latent posterior correlations in a scalable manner. Moreover, AEA-svGPFA is particularly well-suited to settings where posterior inference must be scaled to additional data by changing the inducing locations.
> >
> > - Q4: Additional baselines.
> >
> > We thank the reviewer for the suggestion. We have now provided additional baseline evaluations given vanilla-VAE and SVAE with linear dynamical system prior (SVAE-LDS) models on the synthetic, EEG and population spiking datasets.
> >
> > - Q5: Tractability when order increases.
> >
> > Here for tractability, we assume the recognition potential takes the same tree-structured functional form as the prior tree-structured distribution, but as illustrated in Appendix B of the updated manuscript, the given generative model introduces an additional posterior joint factor over all latent variables. We propose to approach this issue with tensor decomposition techniques (e.g., see Kolda and Bader, 2009) to approximate the high-dimensional tensor representing the joint N-way factor such that exponential complexity is alleviated to linear complexity with respect to the number of latents. Preliminary experiments on the bar dataset show that the approximate joint factor given tensor decomposition yields improvement over tree-structured recognition potential, in turn outperforming standard SVAE and VAE. The tensor decomposition technique could in principle be implemented with arbitrary discrete latent graphical model, e.g., Factorial HMM (Ghahramani and Jordan, 1997). However, we note that a major goal of the current paper is to convey the idea that structured recognition potential works better than mean-field recognition potential under the structured autoencoding framework, and we leave further investigation of instantiation of AEA framework with more complex latent variable models to future work.
> >
> > - Q6: minor comments.
> >
> > -- **Fontsize in Figure 1 and 2.** We thank the reviewer for the suggestion. We have expanded the figures of the graphical models of svGPFA models (Figure 1 in the new manuscript) and the figures of Tree-structured amortised inference (Figure 3 in the appendix of the new manuscript). We hope that the new figures are more readable.
> >
> > -- **Objective for SVAE undefined.** We thank the reviewer for pointing out the current omission. We have now defined the SVAE and AEA objectives in the updated manuscript (see Eq. 5 and the paragraph above Eq. 5). Both objectives are explicitly introduced in the proof of Proposition 3.1 in Appendix D.
> >
> > References:
> >
> > [1] Johnson, M.J., Duvenaud, D.K., Wiltschko, A., Adams, R.P. and Datta, S.R., 2016. Composing graphical models with neural networks for structured representations and fast inference. Advances in neural information processing systems, 29.
> >
> > [2] Lin, W., Hubacher, N. and Khan, M.E., 2018. Variational message passing with structured inference networks. arXiv preprint arXiv:1803.05589.
> >
> > [3] O'keefe, J. and Nadel, L., 1979. Précis of O'Keefe & Nadel's The hippocampus as a cognitive map. Behavioral and Brain Sciences, 2(4), pp.487-494.
> >
> > [4] Yu, Byron M., et al. "Gaussian-process factor analysis for low-dimensional single-trial analysis of neural population activity." Advances in neural information processing systems 21 (2008).
> >
> > [5] Adam, V., Hensman, J. and Sahani, M., 2016, September. Scalable transformed additive signal decomposition by non-conjugate Gaussian process inference. In 2016 IEEE 26th international workshop on machine learning for signal processing (MLSP) (pp. 1-6). IEEE.
> >
> > [6] Duncker, L. and Sahani, M., 2018. Temporal alignment and latent Gaussian process factor inference in population spike trains. Advances in neural information processing systems, 31.
> >
> > [7] Keeley, S., Aoi, M., Yu, Y., Smith, S. and Pillow, J.W., 2020. Identifying signal and noise structure in neural population activity with Gaussian process factor models. Advances in Neural Information Processing Systems, 33, pp.13795-13805.
> >
> > [8] Jensen, K., Kao, T.C., Stone, J. and Hennequin, G., 2021. Scalable Bayesian GPFA with automatic relevance determination and discrete noise models. Advances in Neural Information Processing Systems, 34, pp.10613-10626.
> >
> > [9] Ashman, M., So, J., Tebbutt, W., Fortuin, V., Pearce, M. and Turner, R.E., 2020. Sparse Gaussian process variational autoencoders. arXiv preprint arXiv:2010.10177.
> >
> > [10] Ghahramani, Z. and Jordan, M., 1995. Factorial hidden Markov models. Advances in Neural Information Processing Systems, 8.

---

> ### Author Response · Authors · 2022-08-08
> **Followup on the rebuttal replies**
>
> We wish to follow up on our replies to the reviewer's initial reviews. In particular:
>
> 1. We discussed the significance of the AEA framework, and provided additional implementation on classical latent variable models (Appendix A.2 in the updated manuscript);
> 2. We discussed why the tighter lower bound (Proposition 3.1) is favorable in the perspective of posterior inference (see also Appendix D for further discussion);
> 3. We strengthened the novelty and significance of AEA-svGPFA (relative to existing methods);
> 4. We provided additional baseline comparison on all evaluated datasets (Section 5.1 in the updated manuscript);
> 5. We provided one potential approach for dealing with the tractability issue when the order of dependency increases, but we note that it is not the main objective of the current paper;
>
> We hope our replies and corresponding changes in the updated paper have addressed all concerns/questions raised by the reviewer. If this is indeed the case, we hope the reviewer could consider updating the score accordingly. If not, we hope the reviewer could clarify any remaining concerns/questions and we are happy to engage in further discussion.
>
> We again thank the reviewer for the thoughtful comments and for the time and effort in reviewing the paper.

---

> > ### Comment · Reviewer_D7Ke · 2022-08-09
> > **Thank you for your clarification**
> >
> > I also appreciate the additional experimental results. I have increased my score accordingly.

---

> > > ### Author Response · Authors · 2022-08-09
> > > **Thank you for your reply and for raising the score**
> > >
> > > We wish to thank the reviewer for the reply and for raising the score. We are glad to see that our replies have addressed most of the reviewer's questions/concerns, and that the reviewer appreciate our additional empirical evaluations.

---

### Official Review · Reviewer_L1xL · 2022-07-11

**Rating:** 6
**Confidence:** 3
**Soundness:** 3 good
**Presentation:** 3 good
**Contribution:** 3 good

**Summary:**

The paper proposes a way for amortised inference in structured generative models with explaining away. It defines a specific form of variational posterior that is propositional to the product of the prior distribution and the amortised factor potentials instead of a mean-field ones used in existing work. The paper proves that the proposed inference model gives a tighter ELBO.

It further considers two more structured models with a tree prior or a Gaussian process factor analysis prior and applies the proposed method to construct the variational posterior and perform standard learning as in the original SVAE.

The paper conducts a toy example to show the promise of the proposed method over SVAE and evaluate the proposed model on the EGG data and Population Neuronal Firing Data. It quantitatively compares with SGPVAE.

**Questions:**

See weakness

**Limitations:**

I did not see a negative societal impact.

**Strengths And Weaknesses:**

Strengths
1. The motivation for using a "more structured" posterior is clear and theoretically justified.

2. The presentation is easy to follow and related work is properly discussed.

Weakness
1. I'm not fully convinced by the experiments. It would be much better if the authors can compare the proposed method with SVAE and VMP-SVAE on the commonly adopted models and benchmarks therein. For instance, both SVAE and VMP-SVAE are evaluated with a latent mixture model and a latent state-space model. A direct comparison in such settings can strongly support the motivation and theory.

---

> ### Author Response · Authors · 2022-08-02
> **Thank you for your review, below we address each of your questions/comments.**
>
> We wish to thank the reviewer for the thoughtful comments and suggestions. Below we address each of the mentioned questions/concerns.
>
> - Q1: Direct comparison with SVAE/SIN on latent variable models.
>
> We have now provided a self-contained instantiation of AEA framework with latent Gaussian mixture model (GMM) prior in Appendix A.2 in the updated manuscript. The corresponding empirical evaluation on the pinwheel dataset (Johnson, et al. 2016) and comparison with SIN (Lin, et al. 2018) is shown in Appendix E.2 in the updated manuscript, with accompanying discussion. We only compare with SIN since as claimed in the experimental section in the original paper of SIN, SIN consistently outperforms SVAE on the pinwheel dataset. We are currently working on the implementation of AEA with explicit linear dynamical system (LDS) prior and has not been included in the updated manuscript, which we will include in the final manuscript upon acceptance. We note that we have tested the AEA-svGPFA on the moving dots experiment (Johnson, et al. 2016) and compared with SVAE with LDS prior, and we observe that AEA-svGPFA consistently outperforms SVAE-LDS on the moving-dots dataset, both in terms of reconstruction and generation quality. We do not include this set of results in the paper as we do not think this experiment will support our claim that AEA-svGPFA yields stronger empirical performance due to its ability of scalable learning of a full-covariance variational approximation that captures arbitrary posterior dependency introduced by "explaining away" (whereas the comparison with SGP-VAE (Ashman, et al. 2020) would maximally reflect the effect). However, we are open to suggestions and further discussions from the reviewer, and we will include the discussed results with the moving-dots experiment if the reviewer find including such results would significantly improve the paper.
>
> References:
>
> [1] Johnson, M.J., Duvenaud, D.K., Wiltschko, A., Adams, R.P. and Datta, S.R., 2016. Composing graphical models with neural networks for structured representations and fast inference. Advances in neural information processing systems, 29.
>
> [2] Ashman, M., So, J., Tebbutt, W., Fortuin, V., Pearce, M. and Turner, R.E., 2020. Sparse Gaussian process variational autoencoders. arXiv preprint arXiv:2010.10177.

---

> ### Author Response · Authors · 2022-08-08
> **Followup on the rebuttal replies**
>
> We wish to follow up on our replies to the reviewer's initial reviews. In particular:
>
> 1. We have provided additional instantiation of the AEA framework on latent mixture models (GMM), and empirically evaluated on the pinwheel dataset (Johnson, et al. 2016) and compared with SIN (Lin, et al. 2018), vanilla VAE, and variational GMM (see appendix A.2 and E.2);
> 2. We wish to note that another major component of the paper is the AEA-svGPFA model, which stands as a fully non-linear and fully non-conjugate extension to existing GPFA models, and is the first model that is able to perform scalable amortised variational inference in GP models with full covariance structure (please refer to the updated texts in Section 3.2 in the updated paper for further details);
>
> We hope our replies and corresponding changes in the updated paper have addressed all concerns/questions raised by the reviewer. If this is indeed the case, we hope the reviewer could consider updating the score accordingly. If not, we hope the reviewer could clarify any remaining concerns/questions and we are happy to engage in further discussion.
>
> We again thank the reviewer for the thoughtful comments and for the time and effort in reviewing the paper.

---

> > ### Comment · Reviewer_L1xL · 2022-08-09
> > **Thanks for the feedback**
> >
> > Thanks for the feedback. I'm glad to see the additional results on GMM prior models and update my score.

---

> > > ### Author Response · Authors · 2022-08-09
> > > **Thank you for your reply and for raising the score**
> > >
> > > We wish to thank the reviewer for the reply and for raising the score. We are glad to see that our new experiments on latent-GMM models has improved the soundness of our AEA framework.

---

### Author Response · Authors · 2022-08-02
**General Replies to All Reviewers**

We wish to thank all reviewers for the thoughtful comments and suggestions that help to improve the paper, and we appreciate the shared interests in the AEA-svGPFA model amongst the reviewers. In addition to the individual replies, here we provide a summarised reply to the common concerns/questions raised by the reviewers, and discuss the changes we made to the manuscript relative to the original manuscript.

- Significance/Novelty of Structured Amortised Inference.

We note that the an alternative version of the original SVAE framework could potentially be implemented with structured recognition potential (Johnson, et al. 2016), however this was not explicitly discussed and instantiated in both the original SVAE paper and subsequent papers (e.g., Lin, et al. 2018). Here, with the Tree-structured amortised inference, we concretely instantiate the structured recognition potential, and evaluate the superiority over mean-field recognition potential with structured prior, both theoretically and empirically. We have additionally provided an instantiation of the AEA framework with the latent GMM prior and empirically tested on the pinwheel dataset (Appendix A.2 and E.2 of the updated manuscript). Similar to the experiment on the bar dataset, we observe improvement in terms of free energy, reconstruction MSE and generation quality. We agree that implicitly in the original SVAE paper, structured amortised inference is possible, and here we believe our contribution is to concretely instantiate such idea and prove the advantages of using structured recognition potential under the structured autoencoding framework.

- Significance/Novelty of AEA-svGPFA

AEA-svGPFA supports a fully non-linear, non-conjugate extension to standard GPFA (Yu, et al. 2008, Adam, et al. 2016, Duncker and Sahani, 2018, Keeley, et al. 2020), and is the first model that is able to perform variational inference with full-covariance variational Gaussian approximation that captures intra-latent posterior correlations in a scalable manner. Moreover, with the novel variational approximation form, AEA-svGPFA is particularly well-suited to settings where posterior inference must be scaled to additional data by changing the inducing locations (see the discussion in the last paragraph of Section 3.2 in the updated manuscript).

- Modification to the Manuscript.

We have performed the following modification to the manuscript:

1. We moved the discussion and empirical evaluation on the tree-structured amortised inference to appendix A.1;

2. We provide an additional instantiation of the AEA framework given latent Gaussian mixture model prior, and the accompanying empirical evaluations on the pinwheel dataset (Johnson, et al. 2016) in Appendix A.2 and E.2.

3. We expanded the discussion of the AEA-svGPFA model in Section 3.2 in the updated manuscript, where now we include more details about the derivation.

4. We provided additional discussion in Appendix D on why improved free energy bound lead could be beneficial under the perspective of posterior inference.

5. We provided additional baseline comparison for the experiments in Section 5.

6. We have fixed all mentioned formatting comments (e.g., typos, figure fontsize, etc.) and mixed references in the initial submission;

We appreciate the reviewers for their questions/comments and their time and effort in reviewing the paper. We hope the updates to the manuscript and replies to the corresponding questions resolve the raised concerns hence the reviewers could raise the scores accordingly. We are happy to engage in further discussion if any of the questions/concerns is unresolved or if any new questions might arise.

References:

[1] Johnson, M.J., Duvenaud, D.K., Wiltschko, A., Adams, R.P. and Datta, S.R., 2016. Composing graphical models with neural networks for structured representations and fast inference. Advances in neural information processing systems, 29.

[2] Lin, W., Hubacher, N. and Khan, M.E., 2018. Variational message passing with structured inference networks. arXiv preprint arXiv:1803.05589.

[3] Yu, Byron M., et al. "Gaussian-process factor analysis for low-dimensional single-trial analysis of neural population activity." Advances in neural information processing systems 21 (2008).

[4] Adam, V., Hensman, J. and Sahani, M., 2016. Scalable transformed additive signal decomposition by non-conjugate Gaussian process inference. In 2016 IEEE 26th international workshop on machine learning for signal processing (MLSP) (pp. 1-6). IEEE.

[5] Duncker, L. and Sahani, M., 2018. Temporal alignment and latent Gaussian process factor inference in population spike trains. Advances in neural information processing systems, 31.

[6] Keeley, S., Aoi, M., Yu, Y., Smith, S. and Pillow, J.W., 2020. Identifying signal and noise structure in neural population activity with Gaussian process factor models. Advances in Neural Information Processing Systems, 33, pp.13795-13805.

---

### Author Response · Authors · 2022-08-09
**Thank you to all reviewers during the rebuttal phase**

We wish to thank all reviewers for their time and efforts spent during the review and rebuttal phase. We are glad to see that all reviewers have chosen to increase their scores accordingly, reflecting the reviewers' joint recognition in our replies and in the our (updated) paper.

If there is any additional questions/concerns that might arise, we are always happy to engage in further discussions.

Thank you,
The Authors

---

### Meta-Review · Area_Chair_YipK · 2022-08-30

**Recommendation:** Accept
**Confidence:** Less certain

**Metareview:**

**Summary**: This paper proposes a specific form of variational approximation for amortized inference that is proportional to the product of the prior and amortized factor potentials. This is similar to the approach proposed in work on Structured Variational Autoencoders (Johnson et al, NeurIPS 2016), but differs in that each factor in the prior is paired with a corresponding amortized factor over the same set of variables, whereas the original work on SVAEs assumes mean-field factors over individual variables. The authors evaluate this approach in the context of SVAEs, but the main intended us case is an application to sparse variational Gaussian process factor analysis (svGPFA) models, resulting in a proposed AEA-svGPFA model.

**Strengths**: Reviewer *L1xL* found that this submission proposes a good approach with clear motivation and theoretical justification, with a good presentation and discussion of related work.  Reviewers *D7KE* and *9rkF* also appreciated the novel application of SVAE to the svGPFA model, allowing it to better capture posterior dependencies, whilst retaining the same asymptotic time complexity as SGPVAE, with good results for the AEA-svGPFA on neural spike population data. Reviewer *9rkF* further appreciated the analysis of impact of modeling posterior coorelations.

**Weaknesses**: Reviewers were on the whole not fully convinced by the significance of the application to SVAEs [L1xL, D7KE, 9rKF], and each noted that they felt unqualified to judge the significance of the AEA-svGPFA results due to a lack of familiarity of this model class.

**Reviewer Author Discussion**: The authors updated their submission to add an instantiation of the AEA on latent GMMs, evaluated on the pinwheel domain (Johnson et al. 2016), compared to SIN (Lin et al 2018), a vanilla VAE, and a varaitional GMM. The also discussed why a tighter bound (Proposition 3.1) aids posterior inference (with additional discussion in Appendix D), and provided clarifications to reviewers *L1xL* and *D7KE*. In response to the author discussion, all 3 reviewers raised their score (5->6).

**Reviewer AC Discussion**: While all reviewers indicated to the AC that they are leaning towards acceptance, the are also all in agreement that this a paper that is borderline and could also be rejected. The main point of deliberation remains whether the paper would be stronger if it focused on the the AEA-svGPFA and omitted the AEA-SVAE results (which remain somewhat unconvincing to the reviewers). The reviewers also reiterated that they find it difficult to judge the significance of the AEA-svGPFA results.

**Formatting**: There are minor formatting violations; Table 1 and Figure 2 are about 1 inch too wide, but this is easily addressable.

**Overall AC Recommendation**: The AC took a quick and very cursory look at the the paper. The AC has no immediate concerns about soundness and clarity, but also finds it difficult to assess significance. Taking everything into account, this paper appears narrowly above the threshold for acceptance, but may have to be cut to make room for other submissions.

**Award:**

No

---

### Decision · Program_Chairs · 2022-09-14

Accept